# Maternal Obesity and Kawasaki Disease-like Vasculitis: A New Perspective on Cardiovascular Injury and Inflammatory Response in Offspring Male Mice

**DOI:** 10.3390/nu15173823

**Published:** 2023-08-31

**Authors:** Yuanzheng Zheng, Wenji Wang, Yu Huo, Yonghao Gui

**Affiliations:** 1Cardiovascular Center, Children’s Hospital of Fudan University, Shanghai 201102, China; 2National Health Commission (NHC) Key Laboratory of Neonatal Diseases, Fudan University, Shanghai 201102, China; 3Guangzhou Center for Disease Control and Prevention, Guangzhou 510080, China

**Keywords:** maternal obesity, offspring health, Kawasaki disease, transgenerational effect, inflammatory response

## Abstract

Maternal obesity affects the risk of cardiovascular disease and inflammatory response in offspring. However, the impact of maternal obesity on offspring with Kawasaki disease (KD), the leading cause of childhood acquired heart disease, is still an understudied area. This study aimed to elucidate the impact of maternal obesity on offspring in KD-like vasculitis and the underlying mechanisms. Offspring of obese female mice and normal diet dams were randomly divided into two subgroups. The pups were injected intraperitoneally with either *Candida albicans* water-soluble fraction (CAWS) or phosphate buffered saline (PBS) to establish the obesity (OB)-CAWS group, OB group, wild type (WT)-CAWS group, and WT group. Their weight was monitored during the study. After four weeks, echocardiography was applied to obtain the alternation of cardiac structures. Mouse cytokine panel, Hematoxylin-Eosin (HE) staining, western blot, and real-time qPCR were used to study the pathological changes and protein and RNA expression alternations. Based on the study of pathology, serology and molecular biology, maternal obesity lead to more severe vasculitis and induced altered cardiac structure in the offspring mice and promoted the expression of pro-inflammatory cytokines through activating the NF-κB signaling pathway. Maternal obesity aggravated the inflammatory response of offspring mice in KD-like vasculitis.

## 1. Introduction

During the last decades, the prevalence of maternal overweight and obesity has risen dramatically [1]. Evidence from epidemiological, clinical, and animal studies have demonstrated that obesity affects not only maternal health, but also the long-term outcomes of offspring, with increased risk of obesity and cardiovascular disease (CVD) later in life [2,3,4]. Although the exact mechanisms of the transgenerational effect of maternal obesity remains unclear, fetal programming has been suggested to play a pivotal role in this process [5,6,7]. Human studies have provided evidence for the mechanisms underlying the link between maternal obesity and offspring cardiovascular disease. Offspring of obese mothers have increased body mass index (BMI) and blood pressure, as well as increased carotid artery intima-media thickness (IMT), which is independently linked to the risk of CVD [8]. In addition, animal model studies have also shed light on the mechanisms that drove this linkage [9]. Studies in rodent models have suggested that offspring of obese mothers respond worse to bacterial infection and experimentally induced autoimmunity [10,11,12,13].

Kawasaki disease (KD) is an acute systemic vasculitis that primarily affects young children and results in arterial involvement [14]. The most common sequelae of KD are coronary artery lesions (CAL), which makes it the leading cause of acquired cardiovascular disease in children in developed countries [15]. KD has the highest prevalence in the East Asian population, suggests that genetic factors play an important role in its pathogenesis [16,17]. Despite more than five decades of research, the detailed pathogenesis of KD remains unclear [18]. According to the published studies, inflammatory factors, autoimmune factors, genetic factors, and the interactions between these factors together contribute to the onset of KD [15,19,20,21]. Pathological studies demonstrate that arterial inflammation not only occurs in the acute phase of KD but exists for a much longer period late after the resolution of KD [22]. The chronic inflammation in blood vessels leads to endothelial dysfunction and vascular remodeling, which may further lead to fatal cardiovascular events [23]. Activation of the NF-κB signaling pathway results in the subsequent translocation of NF-κB subunits to the nucleus, where the latter acts as a nuclear transcription factor that promotes the transcription of downstream inflammation-related genes, leading to the occurrence and development of vasculitis [24]. Additionally, studies have shown that the NLR family pyrin domain containing three (NLRP3) inflammasome activation and increased serum levels of proinflammatory cytokines, such as IL-1β, TNF-α and IL-6, also play a critically important role in the development of coronary artery aneurysms and dilatations [18,24].

Since maternal obesity increases the risk of cardiovascular disease in offspring and affects offspring’s response to infection and autoimmunity, the link between maternal obesity and KD is a question worthy of investigation. Although a recent study has shown that obesity is associated with increased odds of total CAL in KD, suggesting that obesity may be an independent risk factor for the development of CAL in KD patients [25]. However, whether the offspring of obese mothers have a higher incidence of CAL in KD, and the relationship between maternal obesity and the severity of KD vasculitis in offspring have not been elucidated.

This study aimed to explore the relationship between maternal obesity and the severity of vasculitis in offspring mice, and the underlying molecular mechanisms during the process.

## 2. Materials and Methods

### 2.1. Ethics Statement

All experimental procedures were performed in accordance with institutional guidelines and approved by the Animal Experiment Committee of the Children’s Hospital of Fudan University (no. 151-2021). Four-week-old C57BL/6 female mice and equal number of male mice were purchased from Charles River Laboratory Animal Technology Co., Ltd. (Jiaxing, China). Mice were maintained in a specific pathogen-free room at 22 ± 2 °C with a 12-h light/12-h dark cycle and provided food and water ad libitum.

### 2.2. Obese Mouse Model and Animal Grouping

Female mice were randomly divided into a normal diet group and an obese group. Both groups of mice were fed the same way as in our previous study [26]. Briefly, Mice in the control group were fed with a normal diet (18.2 kcal% fat, energy 3.5 kcal/g, P1103F-25, SLACOM, Shanghai, China), while the mice in the obese group were fed a western diet (41 kcal% fat, energy 4.7 kcal/g, D12079B, Research Diets, New Brunswick, NJ, USA). Male mice received standard laboratory chow. Eight weeks later, each female mouse was mated with one male mouse. Male offspring mice from the normal diet group and the obese group were collected.

### 2.3. KD Mouse Model and Animal Grouping

*Candida albicans* (NBRC1385) water-soluble fraction (CAWS) was used to induce vasculitis as we previously described [27]. CAWS was prepared from *C. albicans* strain NBRC1358 (NITE Biological Resource Center, Japan). After *C. albicans* was cultured in C-limiting medium at 400 rpm, 27 °C, for 48 h, an equal volume of ethanol was added, and the mixture was incubated overnight. Then, the precipitate was collected and dissolved in 250 mL of distilled water and mixed with an equal volume of ethanol again. After the incubation of the mixture overnight, the precipitate was collected and dried with acetone to obtain CAWS. Five-week-old male mice offspring from both the normal diet group and the obese group were then randomly divided into two groups. That is, the offspring of normal diet dams were divided into wild type (WT) group (*n* = 6) and WT-CAWS group (*n* = 5). Offspring of obese dams were divided into obesity (OB) group (*n* = 5) and OB-CAWS groups (*n* = 5). Mice in the WT-CAWS group and OB-CAWS group were injected intraperitoneally with CAWS (4 mg/mouse/day) for 5 consecutive days from day 0, while mice in the WT group and OB group were injected with phosphate buffered saline (PBS) (0.1 mL/mouse/day) at the same time. Four weeks post injection, mice were euthanized and blood samples were collected and stored at −80 °C until analysis. Abdominal aortas were fixed with 4% paraformaldehyde for histological examination. Hearts were removed and divided into two parts, one was fixed with 4% paraformaldehyde for histological examination, the other was frozen in liquid nitrogen until required for RNA extraction and protein extraction.

### 2.4. Histological Analysis

The fixed abdominal aortas and cardiac tissues were embedded in paraffin and sectioned. In order to observe the histological changes in the abdominal aorta, coronary arteries, and aortic root in detail, horizontal step sections were made every 20 μm. The sections were stained by Hematoxylin and eosin (H&E) with routine techniques and the severity of the arteritis in each group was evaluated under a light microscope.

### 2.5. Western Blot

Heart tissues were homogenized and lysed in radioimmunoprecipitation assay (RIPA) lysis buffer containing protease and phosphatase inhibitors (Sangon Biotechnology, Shanghai, China) for a total protein extraction. Nuclear and cytoplasmic proteins were obtained with a nuclear and cytoplasmic protein extraction kit (Beyotime Biotechnology, Shanghai, China). Protein content was quantified with a bicinchoninic acid (BCA) protein assay kit (Beyotime Biotechnology, Shanghai, China). Equal amounts of proteins were separated with 10–12% sodium dodecyl sulphate-polyacrylamide gel electrophoresis (SDS-PAGE) gels (Beyotime Biotechnology, Shanghai, China) and then transferred to polyvinylidene fluoride (PVDF) membranes (Merck-Millipore, Darmstadt, Germany). After blocking in Tris buffered saline tween (TBST, 10 mM Tris-base, 100 mM sodium chloride, and 0.1% Tween-20, pH 7.50), buffer containing 5% bovine serum albumin (BSA, Bio-Rad, California, CA, USA) for 1 **h** at room temperature and the membranes were incubated with antibodies against Phosphorylated (p)-p65 NF-κB (1:1000, cat. no. 3033, Cell Signaling Technology, Danvers, MA, USA), Histone H3 (1:1000, cat. no. 0863, Affinity, Changzhou, China), Cleaved caspase-3 (1:1000, Cell Signaling Technology, cat. no. 9661) and β-actin (1:1000, Cell Signaling Technology, cat. no. 4970,) at 4 °C overnight. The membranes were washed with TBST buffer and probed with appropriate secondary antibodies (1:2000, ProteinTech, Chicago, CA, USA) for 1 h at room temperature. The protein bands were detected with the Image Lab software system (Bio-Rad, https://www.bio-rad.com) and the optical density of each band was quantified with ImageJ software (NIH, Bethesda, MD, USA, https://imagej.nih.gov).

### 2.6. RNA Extraction and Quantitative Real Time PCR (RT-qPCR)

Heart tissues harvested from mice were homogenized and lysed in Trizol reagent (Invitrogen Life Technologies, Carlsbad, CA, USA) and total RNA was isolated with the chloroform extraction method. After quantification and qualification by spectrophotometry, total RNA was reversely transcribed with a PrimeScript™ RT Reagent Kit (Takara, Kusatsu, Japan) to obtain cDNA. An RT-qPCR analysis was performed using a Roche 480 Real Time PCR System (Roche, Basel, Switzerland) to detect the mRNA expression levels of each target gene in individual samples. Data were normalized to the control *β*-actin and analyzed by the 2^−ΔΔCT^ method. The primers used in this study are listed in Appendix A.

### 2.7. Measurement of Cytokines and Chemokines

Heart tissue and serum levels of cytokines and chemokines were determined with a Bio-Plex Pro Mouse Cytokine Grp I Panel 23-plex (Luminex Corporation, Austin, TX, USA) using Bio-Plex 200 System (Bio-Rad, Hercules, CA, USA) according to the manufacturer’s guidelines. A total of the following 23 cytokines and chemokines: G-CSF, GM-CSF, Eotaxin, INF-g, IL-1a, IL-1b, IL-2, IL-3, IL-4, IL-5, IL-6, IL-9, IL-10, IL-12p40, IL-12p70, IL-13, IL-17, KC, MCP-1, MIP-1a, MIP-1b, RANTES, and TNF-α were assayed with a single assay according to the kit instructions. Briefly, serum samples and heart tissue protein lysate samples were incubated with the microbeads for 30 min. After the incubation, the detection antibodies were introduced subsequently to the tests and incubated for 30 min. Further, the plates were washed with a wash buffer and the supernatant was removed and Streptavidin-Phycoerythrin was added to each well. Next, the plates were shaken at 850 rpm for 10 min and then washed and the supernatant was removed again. Finally, an assay buffer was added to each well to resuspend the pellet, and then the plate was read in a calibrated Bio-Plex machine. Concentrations of above cytokines and chemokines were analyzed with the Bio-Plex Manager 3.0 software (Bio-Rad, Hercules, CA, USA).

### 2.8. Echocardiography

To assess the effect of maternal obesity on left ventricular (LV) structure and function in the KD offspring mice, transthoracic echocardiography was performed with a Vero 2100 system (VisualSonics, Toronto, ON, Canada). The chamber dimensions and thickness of the LV were measured after mice were anesthetized by inhalation of 2% isoflurane. LV posterior wall thickness (LVPW), interventricular septum (IVS), and LV internal dimension (LVID) were recorded during anesthesia and vital signs such as body temperature and heart rate were continuously monitored during anesthesia. The LV mass was then calculated from these parameters with VisualSonics 1.0.0 software. The above parameters of each group were analyzed in a blind fashion.

### 2.9. Statistical Analysis

This study was statistically analyzed using the SAS statistical software (SAS version 9.4), and the data were visualized and presented using GraphPad Prism 8.0 software. All experiments were repeated at least three times to guarantee robust and unbiased results. In comparing the weight changes over time in each group, the area under the curve (AUC value) was calculated separately for each group, and the AUC values of each group were further subjected to one-way ANOVA. Two-way ANOVA was used when comparing transcript levels, protein analysis results and serum and myocardial multifactor assays between groups. *p* < 0.05 represents a statistically significant difference.

## 3. Results

### 3.1. Maternal Obesity Aggravates CAWS-Induced Vasculitis in Offspring Mice

To investigate the effect of maternal obesity on CAWS-induced KD-like vasculitis in offspring mice, obesity maternal models were first established in female mice via a high-fat diet and then mated with normal male mice. After the obese dams became pregnant and gave birth, their male offspring were randomly divided into two groups, the OB group (*n* = 5) and OB-CAWS group (*n* = 5). Meanwhile, the male offspring of normal diet dams were divided into the WT group (*n* = 6) and WT-CAWS group (*n* = 5). Mice in the four groups were given the corresponding interventions (Figure 1). The OB group still weighed more than the WT group on a normal diet, while mice in both WT-CAWS group and OB-CAWS group showed a remarkable weight loss after the CAWS injection, and their body weight gradually recovered after 2 weeks of the experiment. The weight of mice in the WT-CAWS group returned to normal at the 4th week of the study, while the weight of mice in the OB-CAWS group recovered slowly and was still significantly lower than that of the control group at the end of the study (Figure 2A). There was no significant difference in the heart weight to body weight ratio between different groups (Figure 2B). H&E staining showed that the infiltration of inflammatory cells around the aortic root of the mice in the OB-CAWS group was much more severe than those in the other groups, as was the abdominal aorta. The abdominal aorta wall of mice in the OB-CAWS group showed significant hypertrophy and reconstruction, the structure was unclear, and a large number of inflammatory cells were infiltrated around the blood vessels. Similar pathological changes were seen in the aorta (Figure 2C).

### 3.2. Maternal Obesity Induces Altered Cardiac Structure in Offspring Mice

To determine whether maternal obesity affected the cardiac conformation of offspring mice, echocardiography was used to monitor the relevant parameters. In the present study, maternal obesity demonstrated several structural alternations in the left ventricular (LV) of the offspring. In detail, maternal obesity is associated with thicker left ventricular posterior wall in end-systole (LVPWs) and end-diastole (LVPWd) (Figure 3A,B), thicker interventricular septum in end-systole (IVSs) (Figure 3C) and end-diastole (IVSd) (Figure 3D), decreased left ventricular internal dimension in end-systole (LVIDs) (Figure 3E), and end-diastole (LVIDd) (Figure 3F) in male offspring mice. Additionally, CAWS induced KD-like vasculitis also resulted in some structural changes in the left ventricle of the mice, KD is associated with thinner left ventricular posterior wall in end-diastole (LVPWd) (Figure 3B), thinner interventricular septum in end-diastole (IVSd) (Figure 3D), and lighter left ventricular mass (Figure 3G). However, two-way ANOVA analysis revealed that the interaction between the two factors had no significant effect on the above parameters.

### 3.3. Maternal Obesity Exacerbates Offspring Vasculitis through NF-κB Signaling Pathways

The pathological process of obesity and KD both involve inflammatory response, and NF-κB pathway is an important signaling pathway in mediating inflammatory response. Therefore, we explored the role of this signaling pathway in the intergenerational effect of maternal obesity. In addition, the course of KD also involves apoptosis in cardiovascular tissue, so we also assessed the expression levels of apoptosis-related proteins. Results showed that the expression of phospho-p65 protein in the nuclei of cardiomyocytes of OB-CAWS group was increased compared to other groups. And that the expression levels of cleaved caspase-3 in both the OB-CAWS group and WT-CAWS group were higher than that of the WT group and the OB group. (Figure 4A). Quantitative analysis of protein expression indicated that the expression levels of phosphor-p65 in WT-CAWS group, OB group, and OB-CAWS group were higher than those in WT group. In addition, two-way ANOVA demonstrated that both KD and OB promoted the expression of phosphor-p65, and the interaction between the two variables had no effect on the level of this protein (Figure 4B). Moreover, the expression levels of cleaved caspase-3 in both the WT-CAWS group and OB-CAWS group were higher than that of the WT group and OB group. Two-way ANOVA demonstrated that only KD affected the expression of cleaved caspase-3, while OB and the interaction between OB and KD had no effect on the expression of cleaved caspase-3 (Figure 4C).

### 3.4. Transgenerational Effects of Maternal Obesity on Cytokines and Chemokines of Offspring

To investigate the effect of maternal obesity during pregnancy on the inflammatory response to offspring vasculitis, the levels of multiple cytokines in the serum and myocardial tissue of the offspring mice were analyzed. Consistent with previous analysis, Two-way ANOVA was applied to analyze the effect of interaction between maternal obesity and KD on the levels of inflammatory cytokines. As shown in Table 1, obesity was associated with decreased serum levels of Eotaxin and KC, and elevated serum levels of G-CSF, TNF-α in offspring mice, while KD alone was associated with elevated serum levels of G-CSF, IL-1b, IL-6, RANTES, TNF-α and decreased serum levels of IL-1a, IL-9, IL-12 (p70). Moreover, the two factors of maternal obesity and KD together contributed to the increase in serum G-CSF, IL-12 (p40), RANTES levels and decrease in Eotaxin serum levels. We further detected the levels of inflammatory cytokines and chemokines in myocardial tissue. As shown in Table 2, obesity was associated with higher levels of IL-12 (P40), MIP-1a in myocardial tissue, while KD alone was associated with higher levels of G-CSF, IL-9, MCP-1, MIP-1a, MIP-1b, RANTES, and TNF-α in myocardial tissue. In addition, the two factors together contributed to the elevated levels of G-CSF in myocardial tissue.

The effect of the two variables KD and OB on the fold change value of cytokines and chemokines in serum and myocardium was further analyzed. Compared to the WT group, KD raised serum levels of G-CSF, IL-6, TNF-α, IL-1b, and RANTES by 5.08, 3.05, 1.54, 1.46, and 1.46 times, respectively, and reduced serum levels of IL-9, IL-1a, and IL-12 (p70) by 1.17, 1.32, and 1.35 times, respectively (Figure 5A). Compared with the WT group, OB raised serum levels of G-CSF and TNF-α by 2.88 and 1.69 times, respectively, and reduced serum levels of Eotaxin and KC by 1.53 and 1.67 times, respectively (Figure 5B). The interaction of the two variables, KD and OB, together raised serum levels of RANTES, G-CSF, and IL-12 (p40), and reduced serum levels of Eotaxin (*p* < 0.05) (Figure 5C). Additionally, compared with the WT group, KD raised myocardial tissue levels of MIP-1a, MIP-1b, MCP-1, RANTES, TNF-α, G-CSF, and IL-9 by 5.05, 3.89, 2.96, 2.40, 1.46, 1.45, and 1.28 times, respectively (Figure 5D). Compared with the WT group, OB reduces myocardial tissue levels of MIP-1a and IL-12 (P40) by 1.42 and 1.50 times, respectively (Figure 5E). The interaction of the two variables KD and OB together raised myocardial tissue levels of G-CSF (*p* < 0.05) (Figure 5F).

In order to consolidate the above results, real-time quantitative PCR was further applied to detect the transcription levels of inflammatory cytokines and chemokine-related genes in the myocardial tissue of mice. However, the mRNA levels of the above cytokine and chemokine-related genes were not completely consistent with the protein levels. Maternal obesity resulted in elevated mRNA levels of Eotaxin, G-csf, IL-12β-1 in offspring mice, KD-like vasculitis was associated with elevated mRNA levels of G-csf, IL-1a, IL-1b, IL-12β-1, Cxcl-1, Rantes, Tnf-α, and the two factors together contributed to the elevation of mRNA levels of Eotaxin and IL-12β-1 (Figure 6).

## 4. Discussion

In the present study, maternal obesity was associated with more severe vasculitis and altered cardiac structure in offspring mice in KD-like vasculitis. Interestingly, offspring mice of obese dams showed a remarkable weight loss after the CAWS injection and recovered slower than that of the control dams. Furthermore, maternal obesity promoted the expression of pro-inflammatory cytokines through activating NF-κB signaling pathway. To the best of our knowledge, this is the first study to clarify the impact of maternal obesity on offspring in KD-like vasculitis and the underlying mechanisms.

The aim of the present study was to elucidate the influence of maternal obesity on offspring inflammatory response and the underlying mechanisms in KD-like vasculitis, and further explore the potential role of maternal obesity on offspring inflammatory response in cardiovascular disease from a new perspective. In this study, the maternal obesity mouse model was established with an eight-week high-fat diet, and the offspring KD-like vasculitis mouse model was induced by intraperitoneal injection of CAWS. Vascular pathology, echocardiographic parameters, and levels of inflammatory cytokines and chemokines in serum and myocardial tissue were compared between the offspring of obese dams and those of healthy dams. The results of this study show that maternal obesity has a profound impact on the inflammatory response to CAWS-induced vasculitis of offspring mice, even after they have reached puberty. To our knowledge, this is the first study to investigate the effect of maternal obesity on offspring’s inflammatory response to KD-like vasculitis.

In this study, mice in both the WT-CAWS group and the OB-CAWS group had a marked weight loss after the CAWS injection, and gradually recovered after two weeks of the experiment. However, the weight of the mice in the WT-CAWS group returned to normal at the end of the experiment, while the weight of mice in the OB-CAWS group remained below normal. Additionally, the offspring of obese dams showed more severe vasculitis, suggesting that maternal obesity affected the offspring’s growth and inflammatory response. This is consistent with previous studies, which have demonstrated that maternal obesity affects the long-term health of offspring [13,28,29]. The interaction between maternal body mass index and rapid infant weight gain jointly exacerbate the risk of childhood obesity [30]. Moreover, maternal obesity induces metabolic dysfunction in offspring, and the mechanisms involved decreased expression of genes that were related to adipocyte proliferation and differentiation, and insulin signaling and lipid mobilization [31,32]. Additionally, maternal obesity may program perivascular adipose tissue and associated immune cells in offspring, thereby enhancing the obesity/cardiovascular disease link [33,34].

Although a considerable number of studies have been carried out, there are still many unanswered questions concerning the mechanisms of the transgenerational effect of maternal obesity. The present study focused on understanding the potential mechanisms. Based on the study of serology and molecular biology, increased activation of the NF-κB signaling pathway were found in OB-CAWS mice compared with those in other groups. Proteins in the NF-κB family affect several pivotal physiological processes, including inflammation, proliferation, and cell death [35,36]. As a key mediator in the inflammatory response, activated NF-κB family member protein phosphor-p65 promotes the transcription of hundreds of downstream genes, many of which are pro-inflammatory [37,38]. In this study, the expression level of phospho-p65 protein in the nucleus of cardiovascular tissue of mice in the OB-CAWS group was higher than that in other groups, suggesting the activation of pro-inflammatory genes. Histopathological results showed that there was more inflammatory cell infiltration in the cardiovascular tissue of the mice in the OB-CAWS group. Meanwhile, serological results also showed that the serum levels of pro-inflammatory cytokines, such as G-CSF, IL-6, and TNF-α were higher than those in other groups. This evidences, directly and indirectly, that the inflammatory response in the OB-CAWS mice was more severe. Moreover, these results further support the idea that maternal obesity affects the inflammatory response of offspring.

Moreover, increased expression of the pro-apoptotic protein, the important executive protein in apoptosis process, and cleaved caspase-3 [39] were observed in both the OB-CAWS group and the WT-CAWS group. The LV mass of the above two groups was lighter than that of the other groups. The decrease in LV mass was consistent with the increase in pro-apoptotic proteins, suggesting that the decrease in LV mass in CAWS-induced KD mouse model may be related to the increase in cardiomyocyte death, and exacerbated by maternal obesity.

Consistent with previous studies [40,41,42], altered expressions of multiple inflammatory cytokines and chemokines were observed in this study, which was associated with both obesity and KD-like vasculitis. Among the many cytokines that have altered expression levels, G-CSF was in the spotlight. The level of G-CSF was significantly increased in both serum and myocardial tissue of OB-CAWS mice, suggesting that G-CSF may be a specific factor that promotes the aggravation of inflammation in this transgenerational process. As one of the few cytokines known to control neutrophil production, G-CSF is considered to be the guardian of granulopoiesis [43]. By binding to its cognate receptor on the surface of target cells and activating the intracellular signaling pathways, G-CSF regulates the production, differentiation, and release of neutrophil precursors in the bone marrow (BM) and regulates the function of mature neutrophils at sites of inflammation outside the BM [44]. However, studies have demonstrated that G-CSF acts as a double-edged sword in host immunity that is mediated by neutrophils. It plays the dual role of protection and destruction in different situations. Under normal circumstances, neutrophils are essential for the control of infection by the innate immune system, and when excessively produced, recruited, and activated, they may lead to tissue damage in inflammatory diseases [45]. In the first stage of the KD pathological process, neutrophil-dominated immune cells mediate necrotizing vasculitis leading to full-thickness damage to the vessel wall [15,18]. Furthermore, animal study has demonstrated that maternal obesity exacerbated the infiltration of neutrophil in inflammatory response in the airways of male offspring [46]. Collectively, our findings suggest that maternal obesity may affect the long-term immune response and exacerbate KD-like vasculitis in male offspring mice through G-CSF.

Eotaxin is another chemokine that has attracted our attention in this study. As a potent eosinophil chemoattractant, eotaxin mediates eosinophil recruitment into inflammatory sites by activating eosinophil emergence and migration [47]. Elevation in eotaxin and eosinophil levels occur in allergic reactions, asthma, and inflammatory diseases, including KD [48,49,50,51]. However, the reduction of the eosinophil count is involved in other conditions and the recovery of the eosinophil count may promote the repairment of disease injury. A declined blood eosinophil count was observed in ST-segment elevation myocardial infarction, eosinophil deficiency lead to attenuated anti-inflammatory macrophage polarization, and further enhanced myocardial inflammation, increased scar size, and resulted in the deterioration of myocardial structure and function [52]. IL-5 can facilitate the recovery of cardiac dysfunction post-myocardial infarction by promoting eosinophil accumulation and subsequent macrophage polarization [53]. In the present study, maternal obesity was associated with decreased serum level of eotaxin, and the interaction of the two factors maternal obesity and KD also contributed to a decreased serum level of eotaxin. These results suggest that eotaxin may play an important role in the exacerbation of offspring KD-like vasculitis caused by maternal obesity, and that maternal obesity may affect the immune response of offspring by reducing serum eotaxin level and inhibiting eosinophilic recruitment.

Overall, these data suggest that maternal obesity may have transgenerational impacts on offspring inflammatory response in KD-like vasculitis. Based on this study, moderate fat consumption in maternal diets should be recommended in order to avoid dietary fat during pregnancy.

## 5. Strengths and Limitations

We innovatively explored the effect of maternal obesity on the inflammatory response of male offspring in KD-like vasculitis, and comprehensively analyzed the alternations in the levels of inflammatory cytokines in the serum and myocardium of offspring mice. The results demonstrated that maternal obesity during pregnancy had a long-term impact on offspring cardiovascular system. Even when the offspring were in puberty, maternal obesity still affected their cardiac remodeling and response to inflammation. Furthermore, the present study identified a specific increase in the inflammatory molecule G-CSF in this transgenerational effect. This study sheds new light on the impact of maternal obesity on the long-term health and inflammatory responses of offspring.

The expression of pro-apoptotic proteins was increased in OB-CAWS mice, which was accompanied by increased inflammatory factors in myocadiac tissues and serum, suggesting that the cells undergo a severe inflammatory response at the same time as programmed cell death. Therefore, whether other types of cell death, such as pyroptosis, were involved and what role they played deserves further exploration. In addition, future studies should continue to follow the cardiovascular health of OB-CAWS mice at older ages to determine whether the cardiovascular injury of the offspring mice are permanent. Furthermore, the molecular mechanism of the increase on G-CSF factor in myocardial tissue and serum of the offspring mice is also worthy of continued research.

Oxidative stress is closely related to inflammation and plays a pivotal role in the pathology of Kawasaki disease. There is an excessive production of reactive oxygen species which increases oxidative stress in KD during the acute phase, and it triggers an endless vicious spiral of inflammation reactions. Additionally, although the acute inflammatory reactions and oxidative stress can be rapidly controlled by timely and adequate treatment, chronic inflammation and oxidative stress may exist for a long time, which may impair blood vessels, result in carotid intima-media thickening and arterial stiffening, and further lead to the onset of atherosclerosis. Maternal Obesity has also been reported to be associated with increased oxidative stress in offspring. In our present study, we have not investigated the effect of maternal obesity on offspring oxidative stress in KD-like vasculitis, which is a study limitation and need further research.

Since the long-term effect of maternal obesity has profound public health implications, additional research is needed to obtain a mechanistic understanding how maternal obesity affect the cardiac risk and inflammatory response of offspring, and what maternal interventions during key developmental windows may reduce offspring inflammation dysfunction and cardiac risk.

## Figures and Tables

**Figure 1 nutrients-15-03823-f001:**
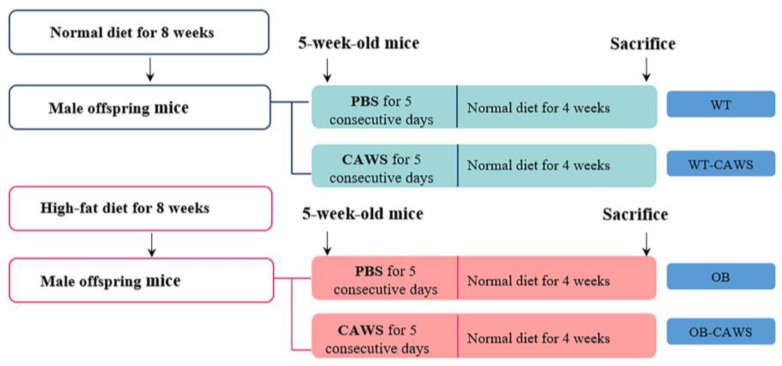
Schematic of the animal model. Obesity maternal model was established in female mice via a high-fat diet and then mated with normal diet male mice. After the obese dams gave birth, their male offspring were randomly divided into two groups, OB group (*n* = 5) and OB-CAWS group (*n* = 5). Meanwhile, the male offspring of normal diet dams were divided into WT group (*n* = 6) and WT-CAWS group (*n* = 5). Mice in the four groups were given corresponding intervention.

**Figure 2 nutrients-15-03823-f002:**
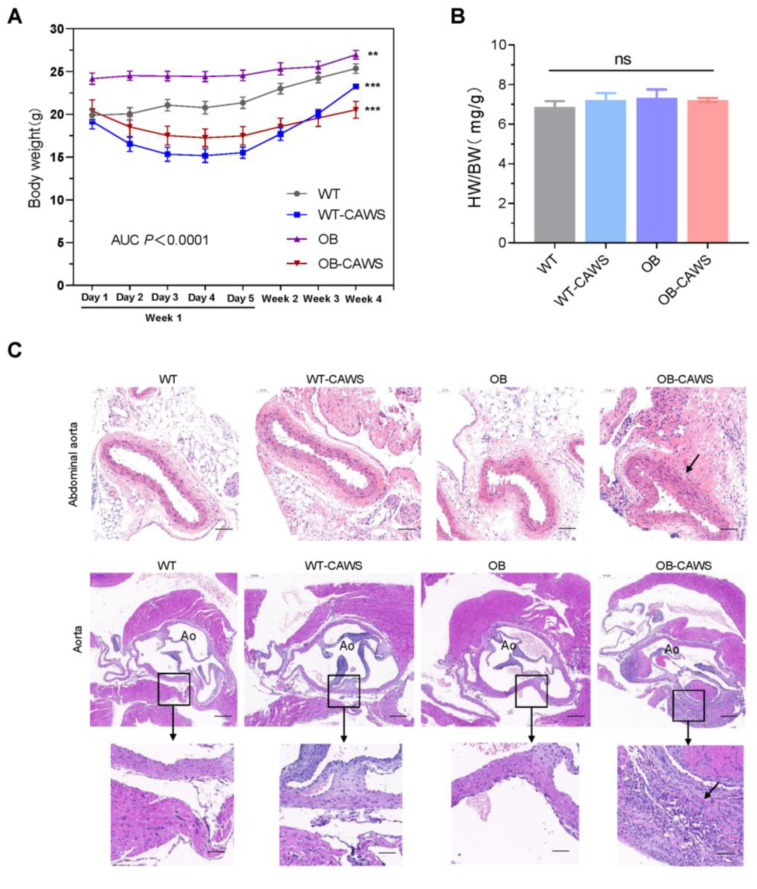
Maternal obesity aggravates CAWS-induced vasculitis in offspring mice. (**A**) OB group still weighed more than the WT group on a normal diet, while mice in both WT-CAWS group and OB-CAWS group showed a remarkable weight loss after the CAWS injection, and their body weight gradually recovered after 2 weeks of the experiment. The weight of mice in the WT-CAWS group returned to normal at the 4th week of the study, while the weight of mice in the OB-CAWS group recovered slowly and was still significantly lower than that of the control group at the end of the study. “**”, “***”, indicates *p* < 0.01, *p* < 0.001, respectively. (**B**) There was no significant difference in the heart weight/body weight ratio between different groups. (**C**) H&E staining was used to examine the histopathological changes of the aorta and abdominal aorta, and representative images are shown. The arrows indicate the main pathological changes. Scale bar of the abdominal aorta: 100 μm. Scale bar of the aorta: scale bars of the upper pictures and the lower pictures are 200 μm and 50 μm, respectively.

**Figure 3 nutrients-15-03823-f003:**
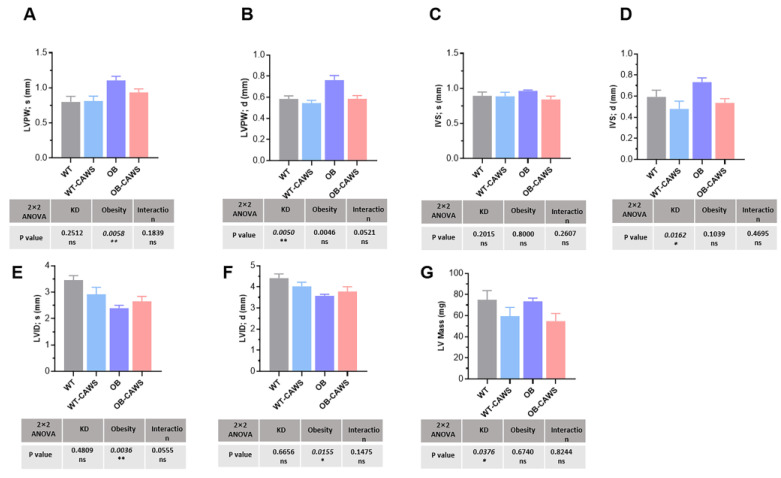
Maternal obesity induces altered cardiac structure in offspring mice. (**A**,**B**) Maternal obesity is associated with thicker left ventricular posterior wall in end-systole (LVPWs) and end-diastole (LVPWd) in offspring mice, KD-like vasculitis is associated with a thinner left ventricular posterior wall in end-diastole (LVPWd). (**C**,**D**) Maternal obesity is associated with thicker interventricular septum in end-systole (IVSs) and end-diastole (IVSd) in offspring mice, KD-like vasculitis is associated with a thinner interventricular septum in end-diastole (IVSd). (**E**,**F**) Maternal obesity is associated with a decreased left ventricular internal dimension in end-systole (LVIDs) and end-diastole (LVIDd) in offspring mice. (**G**) KD-like vasculitis is associated with lighter left ventricular mass. Two-way ANOVA analysis revealed that the interaction between the two factors has no significant effect on the above parameters. ns, not significant, * *p* < 0.05, ** *p* < 0.01.

**Figure 4 nutrients-15-03823-f004:**
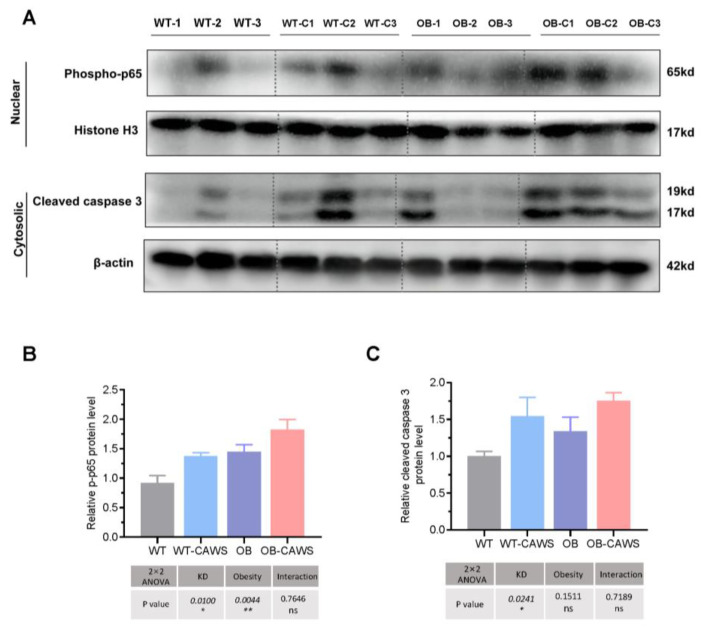
Maternal obesity exacerbates offspring vasculitis through NF-κB signaling pathway. (**A**) The expression of phosphor-p65 protein in the nuclei of cardiomyocytes of OB-CAWS group was increased compared to other groups, and that the expression levels of cleaved caspase-3 in both the OB-CAWS group and WT-CAWS group were higher than that of the WT group and the OB group. (**B**) Quantitative analysis of protein expression indicated that the expression levels of phosphor-p65 in the WT-CAWS group, OB group, and OB-CAWS group were higher than those in WT group. And two-way ANOVA demonstrated that both KD-like vasculitis and OB promoted the expression of phospho-p65, and the interaction between the two variables had no effect on the level of this protein. (**C**) The expression levels of cleaved caspase-3 in both WT-CAWS group and OB-CAWS group were higher than that of the WT group and OB group, two-way ANOVA demonstrated that only KD-like vasculitis affected the expression of cleaved caspase-3, while OB and the interaction between OB and KD-like vasculitis had no effect on the expression of cleaved caspase-3. ns, not significant, * *p* < 0.05, ** *p* < 0.01.

**Figure 5 nutrients-15-03823-f005:**
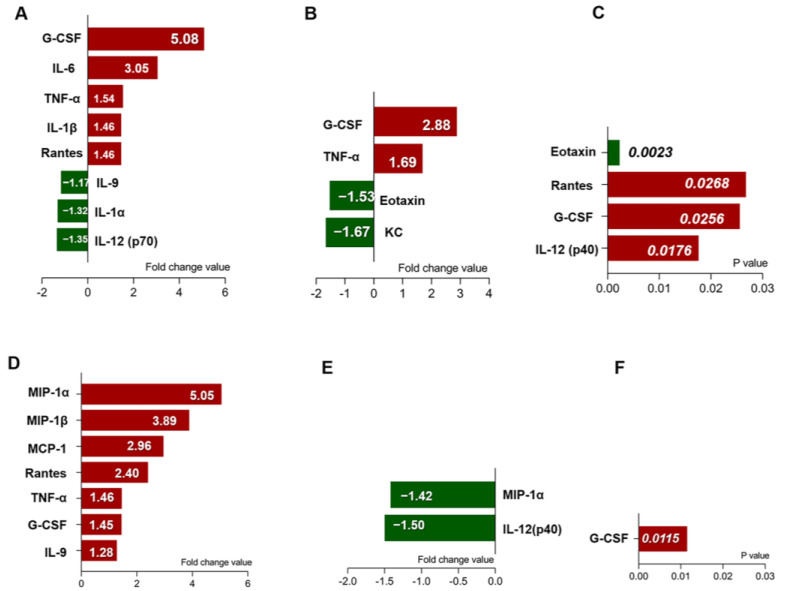
Statistical analysis of the two variables KD and OB on the fold change value of cytokines and chemokines in offspring. (**A**) Compared with the WT group, KD-like vasculitis raised serum levels of G-CSF, IL-6, TNF-α, IL-1b, and RANTES by 5.08, 3.05, 1.54, 1.46, and 1.46 times, respectively, and lowered serum levels of IL-9, IL-1a, and IL-12 (p70) by 1.17, 1.32, and 1.35 times, respectively. (**B**) Compared with the WT group, OB raised serum levels of G-CSF and TNF-α by 2.88 and 1.69 times, respectively, and lowered serum levels of Eotaxin and KC by 1.53 and 1.67 times, respectively. (**C**) The interaction of the two variables KD-like vasculitis and OB together raised serum levels of Rantes, G-CSF, and IL-12 (p40), and lowered serum levels of Eotaxin (*p* < 0.05). (**D**) compared with the WT group, KD-like vasculitis raised myocardial tissue levels of MIP-1a, MIP-1b, MCP-1, RANTES, TNF-α, G-CSF, and IL-9 by 5.05, 3.89, 2.96, 2.40, 1.46, 1.45, and 1.28 times, respectively. (**E**) Compared with the WT group, OB lowered myocardial tissue levels of MIP-1a and IL-12 (P40) by 1.42 and 1.50 times, respectively. (**F**) The interaction of the two variables KD-like vasculitis and OB together raised myocardial tissue levels of G-CSF (*p* < 0.05).

**Figure 6 nutrients-15-03823-f006:**
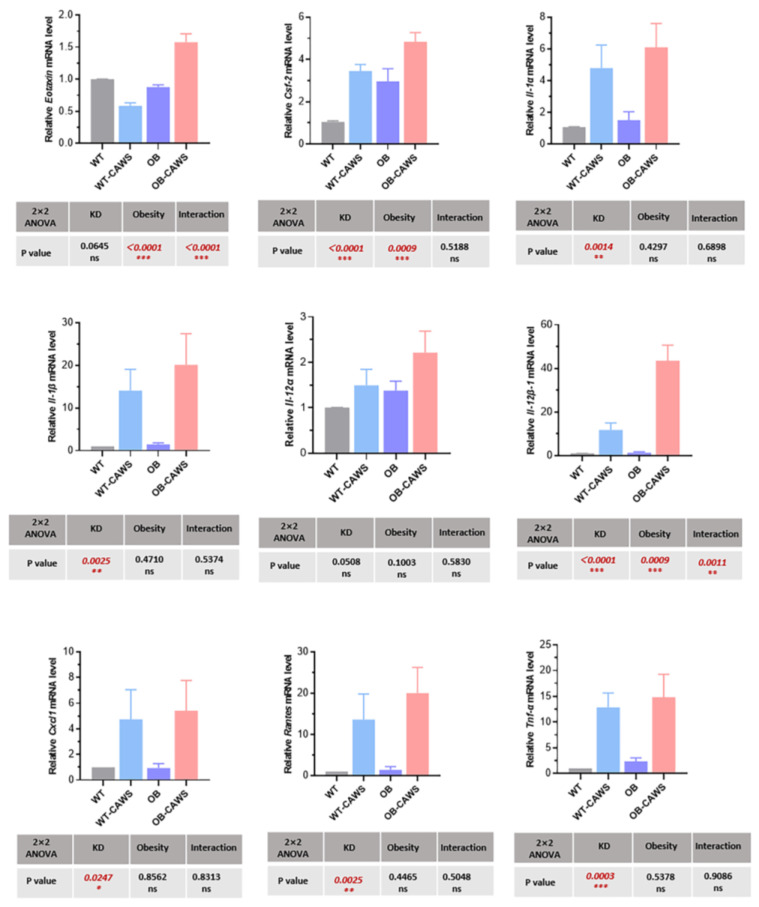
The effects of maternal obesity on mRNA levels of cytokines and chemokines in offspring. Maternal obesity resulted in elevated mRNA levels of *Eotaxin*, *G-csf*, *IL-12β-1* in offspring mice, KD-like vasculitis was associated with elevated mRNA levels of *G-csf*, *IL-1a*, *IL-1b*, *IL-12β-1*, *Cxcl-1*, *Rantes*, *Tnf-α*, and the two factors together contributed to the elevated mRNA levels of *Eotaxin* and *IL-12β-1*. ns, not significant, * *p* < 0.05, ** *p* < 0.01, *** *p* < 0.001.

**Table 1 nutrients-15-03823-t001:** Serum levels of inflammatory cytokines and chemokines in mice.

	Group (Mean ± SD)				*p*-Value (Two-Way ANOVA)		
Variables	WT	WT-CAWS	OB	OB-CAWS	KD	OB	Interaction
Eotaxin (pg/mL)	1840 ± 473.0	983.1 ± 313.5	699.5 ± 104.7	1141 ± 346.9	0.2410	0.0129 *	0.0023 **
G-CSF (pg/mL)	118.5 ± 32.21	324.9 ± 127.4	164.6 ± 29.88	1114 ± 568.0	0.0019 **	0.0143 *	0.0256 *
GM-CSF (pg/mL)	105.9 ± 16.22	101.9 ± 5.014	95.47 ± 7.042	102.9 ± 18.57	0.7971	0.4861	0.3996
IFN-g (pg/mL)	31.49 ± 7.096	27.02 ± 5.440	28.04 ± 1.386	28.36 ± 5.864	0.4568	0.7024	0.3912
IL-1a (pg/mL)	23.95 ± 8.032	17.81 ± 1.085	28.19 ± 3.511	21.81 ± 2.555	0.0186 *	0.0984	0.9588
IL-1b (pg/mL)	8.693 ± 1.382	10.90 ± 2.122	7.065 ± 0.785	12.18 ± 4.718	0.0191 *	0.8992	0.3024
IL-2 (pg/mL)	18.95 ± 3.053	18.86 ± 6.524	14.95 ± 1.473	21.66 ± 3.405	0.1284	0.7740	0.1193
IL-3 (pg/mL)	10.05 ± 1.499	9.215 ± 1.140	11.84 ± 1.777	9.743 ± 1.578	0.0773	0.1511	0.4186
IL-4 (pg/mL)	13.62 ± 1.179	11.18 ± 0.675	17.15 ± 5.582	13.43 ± 2.432	0.0718	0.0884	0.6888
IL-5 (pg/mL)	23.19 ± 3.992	23.19 ± 3.798	19.13 ± 0.790	22.15 ± 6.978	0.5108	0.2745	0.5114
IL-6 (pg/mL)	17.40 ± 2.250	40.11 ± 19.04	15.93 ± 0.472	61.67 ± 35.08	0.0050 **	0.3348	0.2718
IL-9 (pg/mL)	90.26 ± 11.13	76.99 ± 5.113	85.88 ± 7.074	73.01 ± 11.09	0.0131 *	0.3704	0.9652
IL-10 (pg/mL)	163.8 ± 51.58	144.4 ± 12.49	149.9 ± 7.942	126.7 ± 7.674	0.4120	0.2667	0.8992
IL-12 (P40) (pg/mL)	1876 ± 759.4	1559 ± 347.1	1125 ± 110.6	2189 ± 545.9	0.1627	0.8147	0.0176 *
IL-12 (P70) (pg/mL)	467.8 ± 110.8	393.0 ± 51.12	510.7 ± 30.79	331.4 ± 25.70	0.0028 *	0.7866	0.1502
IL-13 (pg/mL)	157.2 ± 24.59	151.1 ± 25.99	125.1 ± 7.921	213.7 ± 85.61	0.1081	0.5230	0.0648
IL-17A (pg/mL)	121.6 ± 57.88	101.4 ± 23.91	133.1 ± 16.24	95.19 ± 30.26	0.1294	0.8844	0.6304
KC (pg/mL)	140.2 ± 12.22	124.1 ± 28.39	77.39 ± 6.945	80.95 ± 21.25	0.5228	0.0001 **	0.3226
MCP-1 (pg/mL)	614.2 ± 80.10	543.9 ± 36.54	518.8 ± 27.17	544.4 ± 81.61	0.4820	0.1494	0.1452
MIP-1a (pg/mL)	6.918 ± 1.239	5.950 ± 0.642	6.350 ± 0.114	7.370 ± 1.344	0.9578	0.3971	0.0632
MIP-1b (pg/mL)	244.6 ± 46.02	230.1 ± 13.72	238.6 ± 16.00	290.5 ± 57.56	0.3496	0.1813	0.1089
RANTES (pg/mL)	324.8 ± 71.44	342.5 ± 67.87	210.6 ± 31.95	440.9 ± 133.0	0.0123 *	0.8546	0.0268 *
TNF-a (pg/mL)	119.7 ± 40.79	105.7 ± 22.20	116.8 ± 8.292	124.3 ± 37.14	0.0149 *	0.0005 **	0.6012

The Serum levels of inflammatory cytokines and chemokines of mice in different groups are shown in the above table. Two-way ANOVA was applied to analyze the effect of interaction between maternal obesity and KD on the levels of inflammatory cytokines. * *p* < 0.05, ** *p* < 0.01.

**Table 2 nutrients-15-03823-t002:** Levels of inflammatory cytokines and chemokines in myocardial tissue.

	Group (Mean ± SD)				*p*-Value (Two-Way ANOVA)		
Variables	WT	WT-CAWS	OB	OB-CAWS	KD	OB	Interaction
Eotaxin (pg/mL)	6.790 ± 2.093	7.328 ± 2.661	5.313 ± 1.044	8.943 ± 3.311	0.1113	0.9557	0.2263
G-CSF (pg/mL)	2.635 ± 0.689	3.470 ± 0.500	3.303 ± 0.688	4.548 ± 1.082	0.0009 **	0.1464	0.0115
GM-CSF (pg/mL)	12.67 ± 1.716	13.84 ± 3.106	11.31 ± 1.345	12.73 ± 4.083	0.3720	0.3939	0.9328
IFN-g (pg/mL)	7.313 ± 1.299	9.198 ± 2.331	6.843 ± 1.756	7.895 ± 0.800	0.3028	0.3028	0.6223
IL-1a (pg/mL)	6.943 ± 0.772	7.718 ± 2.671	8.763 ± 1.345	7.830 ± 2.158	0.9348	0.3257	0.3830
IL-1b (pg/mL)	1.720 ± 0.161	1.963 ± 0.305	1.510 ± 0.352	1.925 ± 0.414	0.0640	0.4573	0.6022
IL-2 (pg/mL)	13.88 ± 1.652	16.45 ± 5.196	18.70 ± 2.447	17.39 ± 3.326	0.7195	0.1184	0.2790
IL-3 (pg/mL)	0.885 ± 0.281	0.903 ± 0.109	6.433 ± 11.71	0.790 ± 0.243	0.4119	0.1502	0.4876
IL-4 (pg/mL)	1.163 ± 0.189	1.255 ± 0.359	0.972 ± 0.493	0.880 ± 0.317	0.1395	1.0000	0.6137
IL-5 (pg/mL)	1.188 ± 0.331	1.128 ± 0.591	1.065 ± 0.510	1.070 ± 0.300	0.7416	0.9196	0.9050
IL-6 (pg/mL)	1.815 ± 1.233	3.043 ± 0.832	2.443 ± 1.314	3.325 ± 1.234	0.0962	0.4513	0.7729
IL-9 (pg/mL)	14.26 ± 4.443	20.08 ± 3.839	15.26 ± 2.311	17.64 ± 2.606	0.0335 *	0.6795	0.3328
IL-10 (pg/mL)	4.150 ± 4.327	7.105 ± 1.084	3.738 ± 1.691	6.685 ± 2.607	0.0507	0.7647	0.9978
IL-12 (P40) (pg/mL)	6.820 ± 1.450	26.41 ± 18.36	6.250 ± 1.062	15.95 ± 3.511	0.2628	0.0089 *	0.3128
IL-12 (P70) (pg/mL)	33.58 ± 5.116	36.95 ± 8.059	26.17 ± 12.44	30.64 ± 2.953	0.3456	0.1110	0.8924
IL-13 (pg/mL)	53.74 ± 25.18	74.52 ± 16.67	57.30 ± 15.76	59.03 ± 7.316	0.2209	0.5067	0.2958
IL-17A (pg/mL)	1.828 ± 0.495	2.000 ± 0.232	1.580 ± 0.507	1.848 ± 0.259	0.2872	0.3313	0.8140
KC (pg/mL)	9.075 ± 1.413	9.643 ± 0.545	8.060 ± 2.345	10.69 ± 2.804	0.1323	0.9882	0.3183
MCP-1 (pg/mL)	32.79 ± 15.21	77.80 ± 41.88	8.393 ± 14.54	75.98 ± 67.69	0.0353 *	0.8295	0.6364
MIP-1a (pg/mL)	0.722 ± 0.121	4.223 ± 1.115	0.572 ± 0.153	2.555 ± 1.029	0.0001 **	0.0346 *	0.0693
MIP-1b (pg/mL)	5.147 ± 4.156	24.37 ± 13.09	5.890 ± 2.969	18.51 ± 10.98	0.0120 *	0.6333	0.5398
RANTES (pg/mL)	41.01 ± 6.148	75.05 ± 20.95	45.88 ± 8.427	128.2 ± 63.89	0.0051 **	0.1139	0.1815
TNF-a (pg/mL)	6.950 ± 3.140	9.090 ± 0.712	6.670 ± 1.612	10.79 ± 1.412	0.0071 **	0.4777	0.3267

The levels of inflammatory cytokines and chemokines in myocardial tissue were shown in the above table. Two-way ANOVA was applied to analyze the effect of interaction between maternal obesity and KD on the levels of inflammatory cytokines. * *p* < 0.05, ** *p* < 0.01.

## Data Availability

Not applicable.

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
