# Peer review of "Maternal Obesity and Kawasaki Disease-like Vasculitis: A New Perspective on Cardiovascular Injury and Inflammatory Response in Offspring Male Mice"

_nutrients, 2023, doi:10.3390/nu15173823_

Round 1
Reviewer 1 Report
The study by Zheng et al. investigated the role of maternal obesity on offspring in Kawasaki –like vasculitis in mice. The disease was induced by the application of Candida albicans. The cardiac function and activity of vasculitis were compared with control mice. The authors studied the underlying molecular mechanisms by measuring protein and RNA expression of the cytokines and markers of inflammation in the heart tissues and serum. The authors concluded that maternal obesity exacerbates offspring vasculitis through NF-κB signaling pathway.
An interesting study that indeed contributes to our knowledge of maternal obesity and cardiovascular diseases.
Minor comments
1. Please give more information for readers about the model.
2. The blood vessels exhibit vasculitis, which is demonstrated in histological slices. The Figure 2C should be improved: Please provide magnification and indicate with the arrows main pathological changes. Give more detailed explanation to HE histology in the text. Did you find a dilatation of the vessels or any changes in media/intima ratio?
Reviewer 2 Report
The present paper aimed to clarify the impact of maternal obesity on offspring in Kawasaki Disease-like vasculitis and the underlying mechanisms.
A few changes are needed, as follows:
Please explain every abbreviation before using it, starting with the abstract.
Figure 4 A: Please explain WT-1, WT-2, WT-3 and so on.
Figure 5: Eotaxin is mentioned. Please add more words about serum Eotaxin within the manuscript.
Discussion should start with your findings, emphasizing what is new in your study.
Kawasaki disease, the most common acquired heart disease in children, an acute, febrile, self-limiting vasculitis, of unknown etiology and no specific diagnostic tests, with a high incidence in Eastern Asia, was associated with an increased oxidative stress, positively correlated with carotid intima-media thickening and arterial stiffening. Please mention as a study limitation missing information about oxidative stress and arterial stiffness in your study.
Which are the implications of your study for clinical practice?
Reviewer 3 Report
For this manuscript the authors are interested in assessing the effect of maternal obesity and Kawasaki disease on cardiovascular disease and inflammatory response of offspring and the underlying mechanisms.
Based on pathological, serology and molecular biology study, the authors concluded that, maternal obesity leaded to more severe vasculitis and induced altered cardiac structure in offspring mice, and promoted the expression of pro-inflammatory cytokines through activating NF-κB signaling pathway. Maternal obesity aggravated the inflammatory response of offspring mice in KD-like vasculitis.
The authors developed an interesting study. Methods were performed appropriately and the objective and conclusion are clear. The manuscript is well written and the experiments are well executed.
Just few clarifications:
1. Figure 2 (C) - add scale and panel names (aorta/coronary arteries) to the figure
2. Figure 5 and 6 - poor readability of diagrams on the figure, I recommend presenting it in the form of a table.
